# Microbial population shift and metabolic characterization of silver diamine fluoride treatment failure on dental caries

**Bidisha Paul[1], Maria A. Sierra[1], Fangxi Xu[1], Yasmi O. Crystal[2], Xin Li[1], Deepak Saxena[1]\*, Ryan Richard Ruff** [3,4]\*

**1** Department of Molecular Pathobiology, New York University College of Dentistry, New York, New York, United States of America, **2** Department of Pediatric Dentistry, New York University College of Dentistry, New York, New York, United States of America, **3** Department of Epidemiology and Health Promotion, New York University College of Dentistry, New York, New York, United States of America, **4** New York University College of Global Public Health, New York, New York, United States of America

\* ds100@nyu.edu (DS); ryan.ruff@nyu.edu (RRR)

**Data Availability Statement:** BIOM files generated with QIIME2 and scripts used to generate figures are available on GitHub (https://github.com/mariaasierra/Dental_Caries).

## Abstract

The objective of this pilot study was to describe the microbial profiles present in the plaque and saliva of children who continued to develop new carious lesions following treatment with silver diamine fluoride ("nonresponders") compared to caries active, caries-free, and children immediately receiving SDF treatment for untreated caries in order to identify potential microbial differences that may relate to a re-incidence of caries. Saliva and plaque samples from infected and contralateral sites were obtained from twenty children who were either caries free, had active carious lesions, were caries active and received SDF treatment immediately before sampling, or had previously received SDF treatment and developed new caries. In total, 8,057,899 Illumina-generated sequence reads from 60 samples were obtained. Reads were processed using the Quantitative Insights Into Microbial Ecology pipeline. Group differences were assessed using Analysis of Variance Models and Tukey Honest Significant Differences. To identify significant taxa between treatment groups, Linear discriminant analysis Effect Size (LefSe) and Analysis of Differential Abundance Taking Sample Variation Into Account were used. Differential abundant analysis indicated that members of the *Lachnospiraceae* family were significantly enriched in non-responders and the genus *Tannerella* and species *Granulicatella adiances* were also highly abundant in this group. LefSe analysis between non-responders and SDF-treated groups revealed that genera *Leptotrichia* and *Granulicatella* were enriched in non-responders. We observed the highest abundance of phosphotransferase system and lowest abundance of lipopolysaccharide synthesis in non-responders. The microbiome in dental biofilms is responsible for initiation and progression of dental caries. SDF has been shown to be effective in arresting the progression carious lesions, in part due to its antimicrobial properties. Findings suggest that the differential abundance of select microbiota and specific pathway functioning in individuals that present with recurrent decay after SDF treatment may contribute to a potential failure of silver diamine

**Funding:** This study was funded in part by awards from the New York University Grant Support Initiative (#RA633, Ruff PI) and the National Institute of Dental and Craniofacial Research (#R56DE028933, Ruff and Saxena, PIs). The content is solely the responsibility of the authors and does not necessarily reflect the official views of the National Institutes of Health, New York University, the New York University College of Dentistry, or the New York University School of Medicine.

**Competing interests:** The authors have declared that no competing interests exist.

fluoride to arrest dental caries. However, the short duration of sample collection following SDF application and the small sample size emphasize the need for further data and additional analysis.

## Introduction

Dental caries is a multifactorial disease that occurs when a dysbiosis of the oral microbiome, fueled by frequent consumption fermentable carbohydrates, favors the proliferation of acidogenic and aciduric microorganisms in susceptible hosts [1]. Known common species that are associated with dental caries include *Streptococcus mutans*, *Streptococcus sobrinus*, and *Veillonella parvula* [2]. The increased prevalence of this taxa leads to a high abundance of organic acids in contact with tooth surfaces, potentially causing the demineralization of enamel crystals [3]. Over time, sustained mineral loss results in the cavitation of enamel and dentinal tissues which if untreated can progress into the pulp of the tooth causing infection, pain, and eventually tooth loss [4,5].

Epidemiologically, caries is the most prevalent childhood disease worldwide [6,7]. Untreated dental caries affects more than 20% of elementary school-aged children in the United States, and over 50% of children have had caries experience [8,9]. For low-income and minority children, caries incidences can exceed 70% and the prevalence of untreated caries is over 30%, which negatively affects the quality of life of this vulnerable population. Children most at risk for oral diseases face profound health inequalities and often lack access to oral health services [10–14]. To address this unmet need, several clinical trials on the pragmatic use of non-surgical methods for caries management are currently in progress [15,16].

Silver diamine fluoride (SDF) is a novel therapy for non-surgical caries management in the United States [17] consisting of a clear liquid that is easily and painlessly applied to caries lesions to arrest their progression. A 38% concentrated SDF application consists of 24–27% silver, 7.5–11% ammonia, and 5–6% fluoride, combining the antibacterial action of silver with the remineralization of fluoride in an alkaline solution that is unfavorable to collagen degradation and has been shown to act as a bactericidal agent to cariogenic bacteria, specifically *Streptococcus mutans* [18]. In vitro studies using single or multi-species biofilm models have shown that silver interacts with bacterial cell membranes and bacterial enzymes to inhibit bacterial growth [19]; however, these antibacterial actions have not yet been confirmed with in-vivo studies. Clinical trials [20] report arrest rates of approximately 80% on primary teeth, with variation in range depending on tooth location, size of the cavity, and presence of plaque. Thus, while effective in the majority of cases, there are lesions for which progression will not be halted by the application of SDF. This treatment failure could be attributed in part to the specific oral microbiome in patients who do not respond to treatment ("non-responders").

The mechanisms underlying arrest failure or refractory growth of caries initially arrested by SDF remains unknown. Our main objective in this study is to analyze the microbial profiles present in the plaque and saliva of children who have developed new carious lesions following SDF treatment. Non-responsive children were compared with those with active caries, those with active caries treated with SDF, and those who were caries-free to identify key taxa that may contribute to a lack of response to SDF therapy.

## Methods

### Study design

This cross-sectional observational pilot study was approved by the NYU School of Medicine IRB (i19-00692). Children attending the pediatric dentistry clinic at the NYU College of Dentistry for periodic preventive or restorative appointments who were between 6 and 13 years old, healthy, and fit study group criteria were approached to participate. Excluded subjects were those who were currently or had been on antibiotic therapy in the previous two months, reported systemic disease in the previous two months, or exhibited uncooperative behaviors that prohibited sample collection.

Consent was collected from the parents of 20 children and child assent was obtained prior to enrollment. Children were divided into four groups of five subjects each according to their clinical profile at enrollment: caries-free, consisting of those who did not present with dental caries; caries-active, those who had active untreated dental caries; SDF-treated, those who presented with open cavities treated with SDF immediately after enrollment; and non-responders, those who had received SDF for dental caries in the past six months but presented with new active caries at enrollment. Study subjects were recruited during their routine scheduled appointments and samples were collected at the same visit. Data on the time of the subjects' last meal, the last toothbrushing or toothpaste exposure, and time of day were recorded, as was the date of the previous SDF application for the non-responder group. Age, sex, ethnicity, and caries status were also recorded on de-identified forms.

Untreated new and reoccurring dental caries were defined by subjects presenting with visible cavitated lesions corresponding to an International Caries Detection and Assessment System (ICDAS) score of 3 or higher. Patients received SDF during the enrollment visit if they presented with visual cavitated lesions consisting of an ICDAS score of 4 or higher.

### SDF application

Oral examinations and subsequent SDF application were performed by a single clinician (YC). All treated areas were similar, as patients were older than five and under continuous care at the New York University College of Dentistry clinic. All subjects with caries had small lesions into enamel and dentin and confined to a single surface only. Infection was treated using a standardized amount of SDF (single drop); an average drop of SDF measuring 32.5 mL consists of approximately 1.64 to 1.75 mg of fluoride and 8.08 to 8.71 mg of silver [21].

### Sample collection

Unstimulated saliva samples and two supragingival plaque samples were collected from participants in a single visit by a single calibrated clinical investigator following a set protocol. For unstimulated saliva, children were instructed to place a 10 ml collection tube under their lip and drool into it until approximately 1 ml of saliva was collected. For plaque samples, a sterile curette was used to collect plaque from specific tooth sites and then placed in a vial with TE buffer, avoiding prolonged air exposure (< 10 seconds). Caries-free children had plaque collected from buccal sites on both sides of a maxillary molar. Children with active decay had plaque collected from the same surface as the active caries lesion as well as the contralateral tooth. Those who had been treated with SDF at study visit had plaque collected from the treated carious surface and the contralateral side within 10 minutes following SDF application. Participants previously treated with SDF presenting with active decay provided a plaque sample from the previously treated tooth and the contralateral side. Saliva samples were collected first, followed by the plaque samples. The total collection time was less than 4 minutes with variation

only in the time it took each child to collect 1 ml of saliva. Samples were immediately transported to a freezing unit and stored at -80˚C until ready for further analysis. The total time from collection to storage was less than 5 minutes for each set of samples.

## Total bacterial DNA extraction and 16S rRNA amplicon sequencing

Total bacterial genomic DNA was purified from 20 saliva samples and 40 sub-gingival plaque samples using the QIAamp PowerFecal Kit (Qiagen). Extracted DNA was then quantified for concentration and purity initially by NanoDrop 2000 spectrophotometer (Thermo Scientific), and further verified fluorometrically by Quant-iT PicoGreen dsDNA Assay Kit (Invitrogen) using the SpectraMax M5 microplate reader (Molecular Devices), then stored at -20˚C until further analysis. Library preparation involved amplifying the V3-V4 hypervariable region of the 16S rRNA gene in the saliva and plaque samples in accordance with the Illumina 16S metagenomics protocol (Part #15044223 Rev. B). The DNA samples were diluted or concentrated to 10 ng/μL. They were further amplified by performing PCR using 2× KAPA HiFi HotStart ReadyMix DNA polymerase (KapaBiosystems) and primer set 341F (5′-CCTACGGGNGGC WGCAG-3′) and 805R (5′-GACTACHVGGGTATCTAATCC-3′), each with overhang adapter sequences (IDT). PCR conditions were 95˚C (3 minutes), with 25 cycles of 95˚C (30 seconds), 55˚C (30 seconds), 72˚C (30 seconds), and a final extension at 72˚C (5 minutes). The resultant amplicons were purified using AMPure XP beads. A second PCR was performed with dual indices from the Nextera XT Index Kit (Illumina) and 2× KAPA HiFi HotStart ReadyMix DNA polymerase. Amplification was performed at 95˚C (3 minutes), with 8 cycles of 95˚C (30 seconds), 55˚C (30 seconds), 72˚C (30 seconds), and a final extension at 72˚C (5 minutes). After purification with AMPure XP beads, the samples were quantified again using the Pico-Green assay, and their size was confirmed by agarose gel electrophoresis. Equimolar amounts of each sample were pooled, denatured, and sequenced using the 300-bp paired-end sequencing protocol and Illumina's MiSeq Reagent Kit V3 (600 cycles).

## Analyses of sequence data

In total, 8,057,899 Illumina-generated sequence reads from 60 samples were obtained. Reads were processed using the Quantitative Insights Into Microbial Ecology pipeline v2.0 (QIIME2) with the following parameters: demultiplexed FASTQs from Illumina were imported with *qiime tools import*—type SampleData[PairedEndSequencesWithQuality]—input-format CasavaOneEightSingleLanePerSampleDirFmt; DADA2 was used to denoised reads with *qiime dada2 denoise-paired—p-trunc-len-f 0—p-trunc-len-r 200*; A phylogenetic tree was created with FastTree using MAFFT alignment with *qiime phylogeny align-to-tree-mafft-fasttree*; For taxonomic classification, the HOMD database [22] was used to train the Naïve Bayes classifier to the 16SrRNA v4-v5 region with the primers 341F 5'–CCTACGGGNGGCWGCAG–3' and 806R 3'–GACTACHVGGGTATCTAATCC–5' using *qiime feature-classifier extract-reads—p-trunc-len 200*.

In total, 3,963 amplicon sequence variants (ASVs) were generated. Diversity analyses, comparisons, and plots were generated in R. Samples collected from saliva and plaque were grouped into SDF-treated, non-responders, caries-free, and caries-active study groups. Alpha diversity indices were calculated using Phyloseq v.1.30 [23]. Group differences were assessed using analysis of variance models (aov) and Tukey's honest significant differences test (TukeyHSD) via the Stats package (v3.6.2) with statistical significance set at $p < 0.05$. To compare microbial composition between groups, a principal component analysis (PCA) was generated based on a Unifrac distant matrix, and significance between clusters was calculated with

permutational multivariate analysis of variance (adonis) using distance matrices (vegan package v2.5.6).

To identify significant taxa between treatment groups, we used Linear discriminant analysis Effect Size (LefSe, Galaxy/Hutlab) and analysis of differential abundance taking sample variation into account (ALDEx2) [24,25].

## PICRUSt analysis

Phylogenetic Investigation of Communities by Reconstruction of Unobserved States (PICRUSt) was used to predict metagenome functional content from marker genes (16S rRNA). Illumina-generated sequences reads were processed using QIIME2 with the following parameters: demultiplexed FASTQs from Illumina were imported with *qiime tools import—type SampleData[PairedEndSequencesWithQuality]—input-format CasavaOneEightSingleLanePerSampleDirFmt*; paired ends sequences were joined with qiime *vsearch join-pairs*; and then filtered based on the quality score with *qiime quality-filter q-score-joined*. Filtered sequences were dereplicated using *qiime vsearch dereplicate-sequences*. Dereplicated sequences were then clustered into OTUs using the closed reference OTU picking method. Clustering was performed at 97% identity against the Greengenes v.13.5 97% OTUs reference database with *qiime vsearch cluster-features-closed-reference*. The exported OTU file was then uploaded to Galaxy/Hutlab for PICRUSt analysis: the OTU table was first corrected for multiple 16S copy number, the normalized OTU table was processed by the 'Predict Metagenome' step to obtain metagenome predictions, and a PICRUSt prediction table at KEGG Pathway Hierarchy Level 3 was produced. PCA analysis, the relative abundance of specific pathways comparisons between treatments, and plots were generated in R.

## Results

### Saliva and plaque samples carry different microbial communities

Most alpha diversity indices did not differ significantly among the four groups. α-diversity measures in saliva samples indicated that (Fig 1A) caries-free and SDF-treated subjects had markedly increased richness compared with caries-active and non-responders based on observed taxa and Simpson indices and increased diversity evenness based on the Shannon index.

In plaque-infected samples, alpha diversity indices were slightly higher in the SDF-treated group based on the Shannon and Simpson diversity indices (Fig 1B). In contralateral plaque, the number of observed taxa in non-responders was lower compared to other groups especially when compared to SDF-treated subjects (p = 0.049) (Fig 1C).

Distinct microbial communities in saliva and plaque were confirmed by PCA generated using UniFrac distance metrics (p = 0.0009) (Fig 1D). Each plaque and saliva group also revealed significantly different clusters in the four treatment groups: caries-active, caries-free, non-responders, and SDF-treated (Fig 1E). Alpha diversity within plaque samples was measured by the number of observed taxa and the Shannon and Simpson diversity indices. Contralateral and infected sites were compared in caries-active (Fig 1F), non-responders (Fig 1G), and SDF-treated (Fig 1H) subjects. Interestingly, the infected site in non-responders had a significantly higher number of observed taxa than their contralateral site (p = 0.029). Contralateral and infected sites had overlapping microbial communities in all four treatment groups (Fig 1I). When comparing treatment groups in plaque samples, caries-active and caries-free groups were significantly different (p = 0.003) (Fig 1J); however, no distinct clusters were found when comparing non-responders and SDF-treated subjects (Fig 1K).

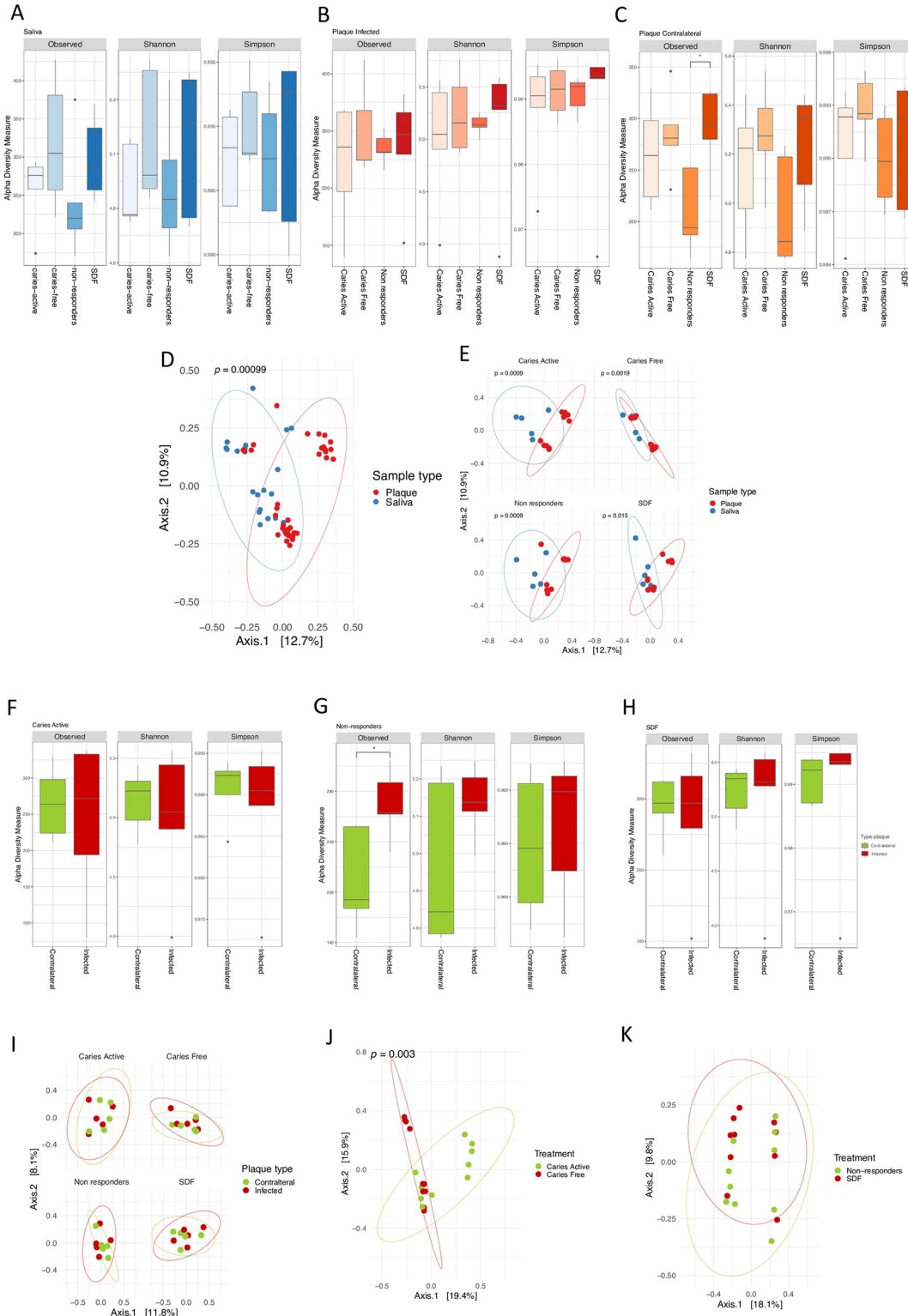

**Fig 1. Microbial diversity varies across saliva and plaque.** Diversity and richness values for (A) saliva samples, (B) plaque infected and (C) contralateral plaque, per treatment group measured by the Observed number of taxa, and the Shannon and Simpson indices. Multiple comparisons of richness and diversity measures were performed by Tukey's range test with *p* values of < 0.05 considered to be statistically significant. Asterisks indicate significant differences between Observed number of taxa in SDF and Non-responders in the contralateral plaque (p = 0.049). (D) Principal Coordinates Analysis (PCoA) based on the

UniFrac distance matrix comparing saliva and plaque samples, and (E) within each treatment group. Comparisons of plaque infected and contralateral site in (F) caries active, (G) non-responders and (H) SDF treated group. (I) Principal Coordinates Analysis (PCoA) based on the UniFrac distance matrix comparing plaque samples types iin each treatment group, (J) comparing caries-active group with caries-free group, and (K) non-responders with SDF treated group.

## Specific taxa might be linked to non-responsive patients

Taxonomic assignment identified the predominance of phylum Saccharibacteria, Firmicutes Bacteroidetes, Proteobacteria, Fusobacteria, and Actinobacteria in plaque-infected samples among all treatments (Fig 2A). Fig 2B depicts the top 40 genera of bacteria in the plaque-infected samples clustered according to the treatment group. Genera commonly of interest in pediatric dental caries research (i.e., *Streptococcus*, *Prevotella*, *Capnocytophaga*, *Leptotrichia*) were found to be highly abundant. At the species level, non-responders seem to have a higher abundance of *Prevotella melaninogenica*, *Veillonella denticariosi, and Rothia dentocariosa*, although no statistically significant differences were observed among the groups. Of note, the dental caries pathogen *Streptococcus mutans* was more abundant in the non-responder and SDF-treated groups compared with the other treatment groups.

LefSe analysis between non-responders and SDF-treated groups in plaque infected samples revealed that the genera *Leptotrichia* and *Granulicatella* were enriched in non-responders (Fig 2C) while the comparison between non-responders and caries-free showed *Saccharibacteria* to be enriched in the caries-free group (Fig 2D). Analyses of differential abundance (ALDEx2) indicated that members of the *Lachnospiraceae* family were significantly enriched in non-responders (Fig 2E) and the genus *Tannerella* (Fig 2F) and species *Granulicatella adiances* (Fig 2G) were also highly abundant in this group. Other group comparisons were not significant.

## Predicted bacterial gene function pathways differential response in treatment groups

Principal Component Analysis (PCA) plot (Fig 3A) does not show any significant difference among groups in plaque infected samples. In PICRUst analyses, we tested pathways that may be relevant in oral microbiome studies of dental caries, such as those related to carbohydrate metabolism, environmental information processing, and human diseases (Table 1). SDF group had lowest abundance of ABC transporters and amino carbohydrate and nucleotide carbohydrate metabolism pathways while having high abundance of bacterial invasion of epithelial cells (Fig 3B). In the environment information processing-membrane transport pathway, the phosphotransferase system (PTS) was more abundant in non-responders. Lipopolysaccharide synthesis was highest in caries-free subjects and lowest in SDF-treated group. However, there were no statistically significant differences observed among the groups in any of the pathways.

## Discussion

Dental caries remains a significant and persistent oral health malady in children. The microbiome in dental biofilms is responsible for the initiation and progression of dental caries. Studies suggest that SDF is effective in arresting the progression of caries lesions due to the antimicrobial properties of silver and fluoride, the hardening of the lesion due to the re-mineralizing effects of fluoride, and the reduction of collagen degradation through inhibition of specific enzymes (metalloproteinases and cathepsins) [31]. However, despite a high efficacy rate, some children or lesions remain resistant to its therapeutic effects. In the present study, we assessed the microbial composition of saliva and plaque of four treatment groups—caries-

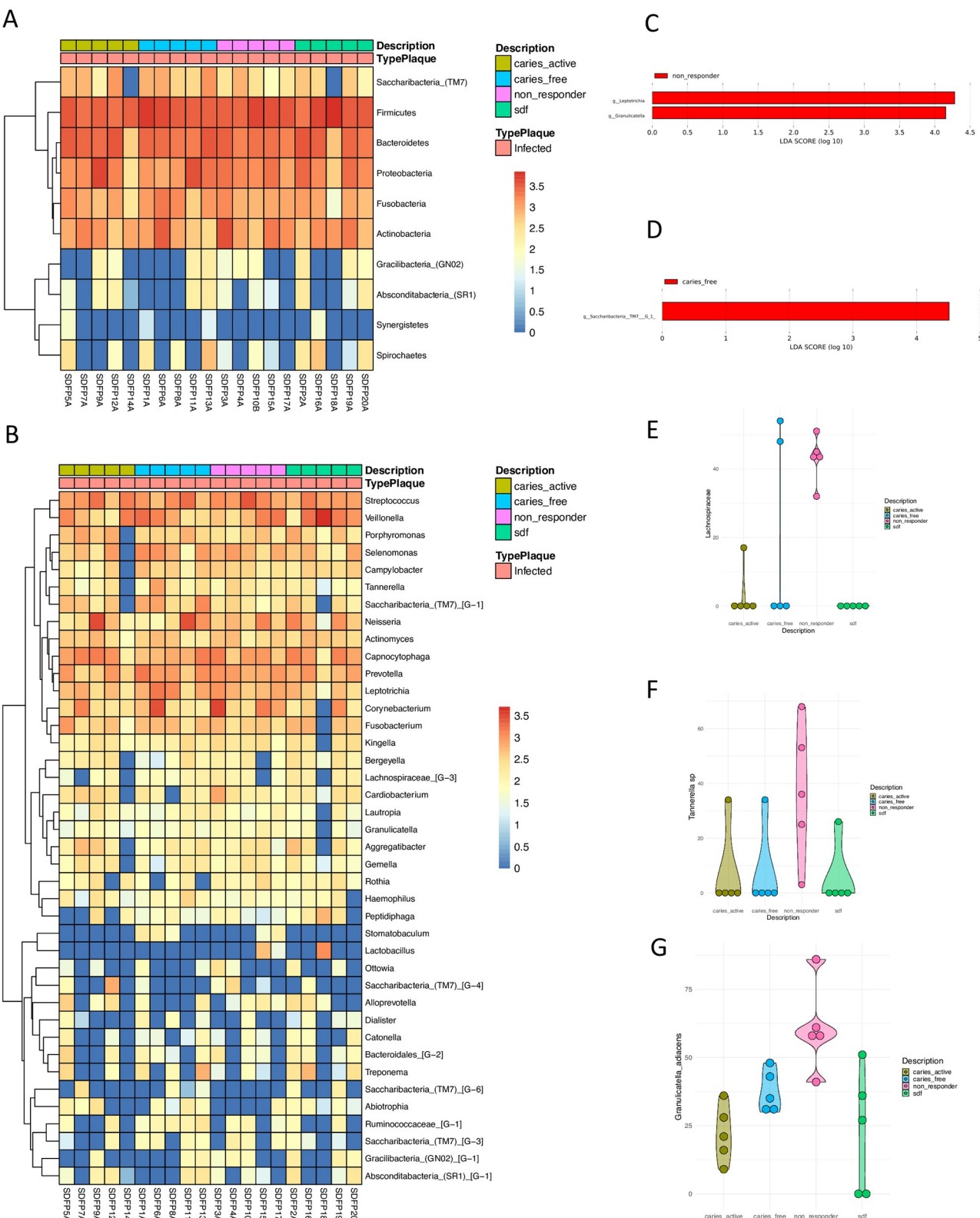

**Fig 2. Microbial shifts across treatment groups.** (A) Heatmap of the relative abundance of taxa present in the infected plaque at phylum level Log10 transformed. (B) Heatmap of the relative abundance of the top 40 most abundant taxa in the infected plaque at genus level Log10 transformed. Significant genera (LEFse) between non-responders and SDF-treated (C) and caries-free (D) groups. Taxa found significantly different in abundance in the four treatment groups of the infected plaque, corresponding to (E) *Lachnospiraceae*, (F) *Tannerella*, and (G) *Granulicatella adiacens*.

A

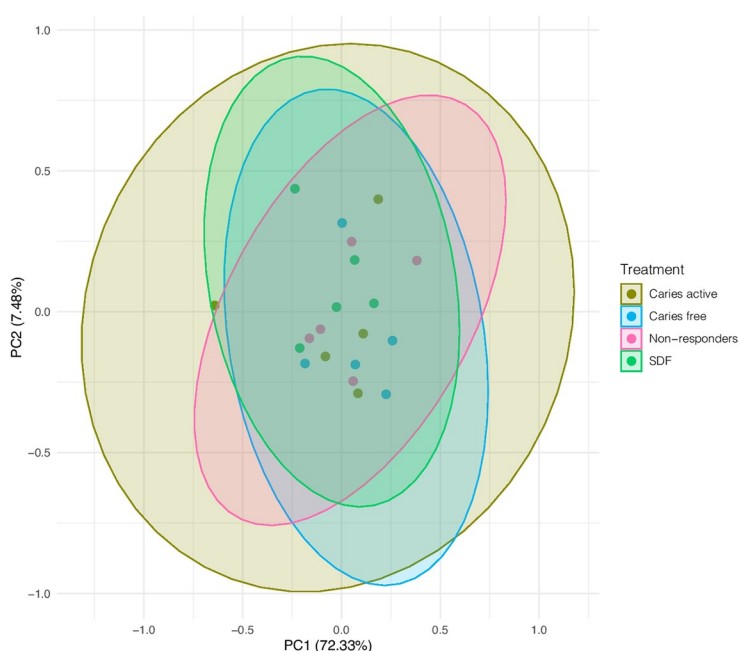

B

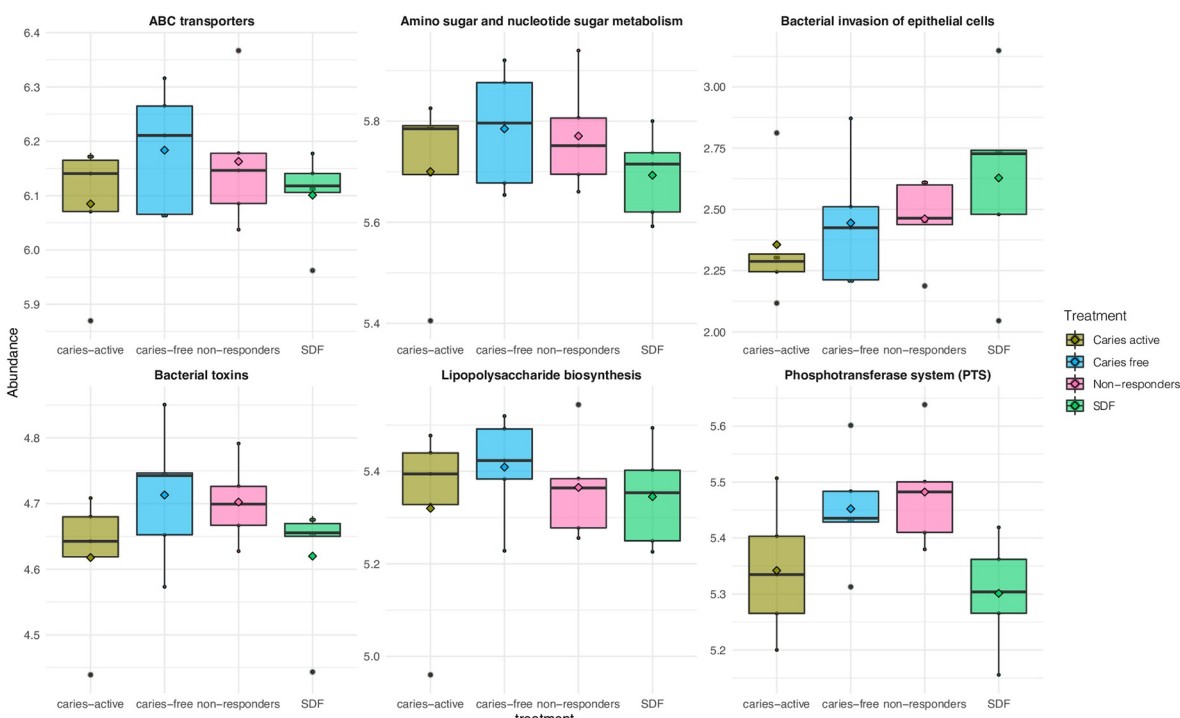

**Fig 3. PCA plot for pathways by group (A); Boxplots for relative abundance of select pathways (B).**

**Table 1. KEGG pathways in oral microbiome studies.**

| Pathways (Level 1,2 and 3) | Importance in oral studies |
|---|---|
| Metabolism; Carbohydrate Metabolism; **Amino carbohydrate and nucleotide carbohydrate metabolism** | Amino carbohydrates are the major components of cell envelopes of fungi and bacteria and they are also important constituents of glycoproteins across all domains of life [26] |
| Environmental Information Processing; Membrane Transport; **ABC transporters** | Maltose transporter (MalEFGK2) predominant in anaerobes with substrate-level phosphorylation [27] |
| Environmental Information Processing; Membrane Transport; **Phosphotransferase system (PTS)** | Glucose transporter (IIAGlc/IICBGlc) Mannose/glucose transporter (IIABMan/IICMan/IIDMan) [27] |
| Human Diseases; Infectious Diseases; **Bacterial invasion of epithelial cells** | Oral pathogenic bacteria adhere to and invade both monolayers and multilayers of primary gingival epithelial cells (GEC) [28]. |
| Environmental Information Processing; Signaling Molecules and Interaction; **Bacterial toxins** | Periodontal pathogens release endotoxin that interacts with the tissues and the host's response [29]. |
| Metabolism; Glycan Biosynthesis and Metabolism; **Lipopolysaccharide(LPS) biosynthesis** | LPS comprising the outer surface of Gram-negative bacteria. High levels of lipid A from *P. gingivalis* indicative of inflammation in periodontal tissues [30]. |

active, caries-free, non-responders, and SDF-treated—and identified bacterial species that are significantly enriched in non-responders.

Overall, there were distinct microbiome profiles among the saliva and plaque samples, supporting results from previous studies that implicate specific microorganisms in the cariogenic process [32,33]. Alpha diversity indices indicated that there were no significant differences across treatment groups, similar to research showing no differences in diversity comparing caries-free and caries-active children [34], however, samples from infected sites had more richness and diversity than contralateral site and saliva. Though not significant, SDF-treated subjects had the highest richness. Comparisons between treatment groups revealed a distinct microbiota of plaque and saliva samples, but as expected, there were no differences in microbial composition between infected and contralateral plaque sites. Interestingly, the non-responder group had a higher number of observed taxa in the infected site compared with its contralateral side.

Although PCA did not reveal any significant differences among the four groups, there was a significant differential abundance of bacteria at the genus and species level detected with ALDEx2. The genus *Granulicatella* was found to be higher in non-responders when compared with the SDF-treated group. *Granulicatella* belongs to the nutritionally variant Streptococci (NVS) group, a group of fastidious, Gram-positive bacteria. NVS require an L-cysteine- or pyridoxal-supplemented medium to support growth and can be distinguished from other streptococci by the presence of cell wall chromophore as well as its distinct enzymatic and serological characteristics [35]. *Granulicatella* is involved in microbial coaggregation especially with *F. nucleatum* which influences the development of complex multi-species biofilm [36]. Since non-responders possess a high abundance of this bacteria, its acidic products of metabolism along with its propensity to aggregate with other colonizers may contribute to resistance against the anti-bacterial properties of SDF. The species *Granulicatella adiacens* that was significantly higher in non-responders is frequently found in dental plaque, endodontic infection, and dental abscesses, and is also associated with other serious infections [37]. By comparing non-responders and SDF-treated subjects using LEfSe, in addition to *Granulicatella*, the genus *Leptotrichia* was more abundant in the plaque-infected site. *Leptotrichia* is Gram-negative, bacilli, anaerobic, non-spore-forming, saccharolytic, and primarily reside in the oral cavity [38]. Previous studies have shown an association between dental caries and *Leptotrichia* and it

likely plays an important role in the caries development process [39–41]. They are non-motile and ferment carbohydrates to produce various organic acids, including lactic acid, and traces of acetic, formic, or succinic acid, depending on the substrates and species. *Leptotrichia* thus possesses characteristics similar to *Streptococcus mutans* as it ferments a large variety of mono- and disaccharides into lactic acid [42]. The acid produced by the bacteria may directly contribute to the demineralization of the tooth enamel, or may attract other aciduric bacteria that contribute to the caries process.

We also report a high abundance of the genus *Tannerella* in non-responders. The most prominent species of the group, *Tannerella forsythia*, is a member of the "Red Complex" group [43]. *T. Forsythia* is the only member of the genus that can be cultured and forms a synergistic biofilm with *F. nucleatum* [44]. The antibacterial activity of silver ions in SDF is lower in dual-species biofilms compared with single-species biofilms [31]. The abundance of *Tannerella* in non-responders may reflect SDF's inability to combat dual-species biofilm formation. However, the presence of the genus *Tannerella* merits more study.

Analysis of predicted microbiota functioning revealed that the phosphotransferase system (PTS) was higher in non-responders. The PTS performs catalytic and regulatory functions in bacteria. It acts as a catalyst for the transportation and phosphorylation of a variety of carbohydrates and carbohydrate derivatives and also regulates the virulence of certain pathogens [45]. Bacteria belonging to genus *Tannerella*, *Leptotrichia*, and *Granulicatella*, which are highly abundant in non-responders, could be utilizing the PTS system to uptake carbohydrates for their metabolism or the expression of their virulence genes and, thus, exhibited an increase in this function.

16S rRNA gene amplicon sequencing captures broad shifts in community diversity over time, but with limited resolution and lower sensitivity compared to metagenomic shotgun sequencing. Our use of the PICRUSt package to predict metagenome functional content from 16S rRNA extends our analysis. Though bias may have been introduced due to varying 16S rRNA PCR amplification frequencies, sequencing errors due to PCR amplification were mitigated by removing mismatches from primers, and denoising and trimming reads with low quality scores during sequencing analyses. Further, the use of a single application of SDF, a relatively short duration between SDF application and sample collection, and the small sample size per group limit generalizability to the greater responder and non-responder populations. As this was a pilot study, access to subjects and corresponding specimens was limited to a single observation. Though identifying the DNA of present bacteria does not elucidate their viability, we elected to investigate the potential immediate measurable effect of therapy to support future study.

Our results are consistent with other research, as no overall significant microbial changes were observed between subjects that had active caries following SDF application and those with caries arrested by SDF [46]. SDF may need to be repeatedly applied at specific intervals to exert its desired effects on the recolonizing bacteria. However, the primary objective of this study was to explore the microbial composition of non-responders that seem to resist the potent antibacterial properties of SDF and how they differ from other groups of patients. The bacteria abundant in non-responders, including *Tannerella*, *Granulicatella*, and *Leptotrichia*, may play a role in a child's lack of response to SDF treatment. In-depth studies of non-responders' biofilm formation mechanisms would likely provide insight into how these organisms evade the antibacterial properties of SDF. Overall, the increased utilization of SDF and other non-surgical interventions for caries in community settings [15,16] can reduce oral health inequities [47], but a lack of treatment response may leave a susceptible portion of the population vulnerable to disease progression. Additional study is needed to support a deeper understanding of the mechanistic actions underlying the non-responsiveness to SDF; future

planned research will follow subjects longitudinally with variable rates of follow-up post-SDF application.

## Author Contributions

**Conceptualization:** Ryan Richard Ruff.

**Data curation:** Yasmi O. Crystal.

**Formal analysis:** Bidisha Paul, Maria A. Sierra, Fangxi Xu, Yasmi O. Crystal, Xin Li, Deepak Saxena, Ryan Richard Ruff.

**Funding acquisition:** Deepak Saxena, Ryan Richard Ruff.

**Project administration:** Ryan Richard Ruff.

**Supervision:** Deepak Saxena, Ryan Richard Ruff.

**Writing – original draft:** Bidisha Paul, Maria A. Sierra, Fangxi Xu, Yasmi O. Crystal, Deepak Saxena, Ryan Richard Ruff.

**Writing – review & editing:** Bidisha Paul, Maria A. Sierra, Fangxi Xu, Yasmi O. Crystal, Xin Li, Deepak Saxena, Ryan Richard Ruff.

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
