## [Decision Letter · Decision Letter 0]

8 Dec 2020

PONE-D-20-34812

Microbial population shift and metabolic characterization of silver diamine fluoride treatment failure on dental caries

PLOS ONE

Dear Dr. Ryan Richard Ruff,

Thank you for submitting your manuscript to PLOS ONE. After careful consideration, we feel that it has merit but does not fully meet PLOS ONE’s publication criteria as it currently stands. Therefore, we invite you to submit a revised version of the manuscript that addresses the points raised during the review process.

Please respond carefully to the reviewer’s comments point by point, especially the rationale for the small sample size and short-term treatment.

We look forward to receiving your revised manuscript.

Kind regards,

Ping Xu

Academic Editor

PLOS ONE

Journal Requirements:

2. Please ensure that the Limitations of the study, such as the y short duration between sample collection and SDF application and the sample size, are mentioned in the Abstract.

"This study was funded in part by an award from the New York University Grant Support Initiative

(#RA633, Ru PI) and the National Institute of Dental and Craniofacial Research (#R56DE028933,

Ru and Saxena, PIs). The content is solely the responsibility of the authors and does not necessarily

re

ect the ocial views of the National Institutes of Health, New York University, the New

York University College of Dentistry, or the New York University School of Medicine."

Reviewers' comments:

Reviewer's Responses to Questions

**Comments to the Author**

1. Is the manuscript technically sound, and do the data support the conclusions?

Reviewer #1: Yes

Reviewer #2: No

2. Has the statistical analysis been performed appropriately and rigorously? 

Reviewer #1: Yes

Reviewer #2: Yes

3. Have the authors made all data underlying the findings in their manuscript fully available?

Reviewer #1: Yes

Reviewer #2: Yes

4. Is the manuscript presented in an intelligible fashion and written in standard English?

Reviewer #1: Yes

Reviewer #2: Yes

5. Review Comments to the Author

Reviewer #1: Please address the following issues:

In the last section of the Introduction, remove metabolites from the text since this was not determined.

Throughout the text ensure that species names are italicized and Gram is written with a capital G.

If available provide more details of the SDF application among the study subjects. Was it applied by the same clinic/person? was a similar size area treated?

Discuss the limitations of the study including the small sample size, problems of 16S sRNA amplification bias.

Change sugar throughout the text to carbohydrate.

Reviewer #2: This manuscript examines the response of oral caries flora to treatment with silver diamine fluoride (SDF), a therapy used for non-surgical caries management in the US. Multiple in vitro studies have shown that SDF inhibits bacterial cell growth through interaction with bacterial cell membranes and enzymes. A primary goal of this study is to examine the efficacy of SDF in a cohort of 20 children 6-13 years of age, and identify the oral bacteria that contribute to a non-responder phenotype, in which the bacteria are resistant to or survive treatment with SDF and have low treatment efficacy. The primary limitations of the study include the small sample size and study design.

Comments:

1. The sample size is small, with 5 children in each of 4 separate treatment groups. The authors do clearly state that this is a pilot study, but do not sufficiently emphasize this limitation when drawing their conclusions in the results and discussion.

2. Is SDF bacteriostatic or bacteriocidal? This is a key, but unaddressed point, in the paper. Why would a bacteriostatic agent be an effective therapeutic?

3. In group 3, the authors sample plaque from surface of a tooth with open caries within 10 minutes after application of SDF. A reasonable rationale for this short time period between SDF application and plaque sampling is needed. An important aspect that the authors do not consider is viability of bacteria on the tooth surface. Taxonomic identification by 166 rRNA amplicon sequencing relies only on the presence of bacterial genomic DNA. It does not access viability of bacteria at that site.

4. The accepted term is 16S rRNA amplicon sequencing.

5. The print (text figure labels) in Figures 1-3 is small and will be very difficult to read when the figures are reduced for publication.

6. It is not clear what Fig 3 (PiCrust) contributes to the paper. None of the identified pathways are statistically significant and no meaningful conclusions can be drawn from this analysis.

7. The identification of Granulicatella, Leptotrrichia, and Tannerella in the non-responders is interesting, with a plausible biological basis of their roles in caries presented in the discussion.

6. PLOS authors have the option to publish the peer review history of their article (what does this mean?). If published, this will include your full peer review and any attached files.

Reviewer #1: No

Reviewer #2: No

---

## [Author Response · Author response to Decision Letter 0]

1 Jan 2021

Thank you for your assistance, we have updated the manuscript to align with the listed style requirements.

2. Please ensure that the Limitations of the study, such as the y short duration between sample collection and SDF application and the sample size, are mentioned in the Abstract.

We have added these limitations to the revised Abstract as appropriate. 

"This study was funded in part by an award from the New York University Grant Support Initiative

(#RA633, Ru PI) and the National Institute of Dental and Craniofacial Research (#R56DE028933,

Ru and Saxena, PIs). The content is solely the responsibility of the authors and does not necessarily

re

ect the ocial views of the National Institutes of Health, New York University, the New

York University College of Dentistry, or the New York University School of Medicine."

We have added the missing funder to the list on the online submission system and removed any funding language from the manuscript. As well, the cover letter is updated as instructed (I could not find the specific funding statement online, only the place to upload the funder information). 

5. Review Comments to the Author

We sincerely thank the two reviewers who graciously offered their time to review our paper and provide feedback. We address each comment as listed below and provide our responses. 

Reviewer #1: Please address the following issues:

In the last section of the Introduction, remove metabolites from the text since this was not determined.

This has been removed

Throughout the text ensure that species names are italicized and Gram is written with a capital G.

This has been checked and updated.

If available provide more details of the SDF application among the study subjects. Was it applied by the same clinic/person? was a similar size area treated?

All SDF treatments and oral examinations were performed by a single clinician (YC).

All areas treated were similar; since patients were older than 5 and under continuous care at our clinic, they had small lesions into enamel and dentin, confined to one dental surface only. Only one tooth was treated at that visit and sampled as described in the methods (sampling also the contralateral tooth, and using a similar amount of SDF (never more than 1 drop). We note a previous publication by the member of our research team who provided treatment on the average amount of F and Ag in a single drop: an average (SD) drop measuring 32.5 mL (microliters) would have approximately 1.64 (0.04) to 1.76 (0.05) mg of fluoride, and 8.08 (0.13) to 8.71 (0.38) mg of silver. We have added this to the manuscript. 

Discuss the limitations of the study including the small sample size, problems of 16S sRNA amplification bias.

We have added additional information (there is a similar comment regarding sample size from the other reviewer) to the discussions/limitations, specifically: 16S rRNA gene amplicon sequencing captures broad shifts in community diversity over time, but with limited resolution and lower sensitivity compared to metagenomic shotgun sequencing. Our use of the PICRUSt package to predict metagenome functional content from 16S rRNA extends our analysis. Additionally, bias may have been introduced due to varying 16S rRNA PCR amplification frequencies. Sequencing errors due to PCR amplification were mitigated by removing mismatches to primers, and denoising and trimming reads with low quality scores during sequencing analyses.

Change sugar throughout the text to carbohydrate.

These have been changed

Reviewer #2: This manuscript examines the response of oral caries flora to treatment with silver diamine fluoride (SDF), a therapy used for non-surgical caries management in the US. Multiple in vitro studies have shown that SDF inhibits bacterial cell growth through interaction with bacterial cell membranes and enzymes. A primary goal of this study is to examine the efficacy of SDF in a cohort of 20 children 6-13 years of age, and identify the oral bacteria that contribute to a non-responder phenotype, in which the bacteria are resistant to or survive treatment with SDF and have low treatment efficacy. The primary limitations of the study include the small sample size and study design.

Comments:

1. The sample size is small, with 5 children in each of 4 separate treatment groups. The authors do clearly state that this is a pilot study, but do not sufficiently emphasize this limitation when drawing their conclusions in the results and discussion.

We agree with the reviewer regarding the limitation of our study. We have added few lines in the results and discussion of the manuscript to address this issue., specifically in the picrust results and in the concluding paragraph of the discussion. This perspective is both from the microbiologic and epidemiologic. 

2. Is SDF bacteriostatic or bacteriocidal? This is a key, but unaddressed point, in the paper. Why would a bacteriostatic agent be an effective therapeutic?

SDF is bactericidal, we have noted this in the introduction to the paper and provided a citation which states that in a review of studies on the effect of SDF on cariogenic bacteria, specifically Streptococcus mutans (Zhao et al, 2018; IDJ DOI: 10.1111/idj.12320). Others include Abdullah et al, 2020; https://doi.org/10.1371/journal.pone.0241519 & Burgers et al, 2009; https://doi.org/10.1016/j.archoralbio.2009.03.004.

3. In group 3, the authors sample plaque from surface of a tooth with open caries within 10 minutes after application of SDF. A reasonable rationale for this short time period between SDF application and plaque sampling is needed. An important aspect that the authors do not consider is viability of bacteria on the tooth surface. Taxonomic identification by 166 rRNA amplicon sequencing relies only on the presence of bacterial genomic DNA. It does not access viability of bacteria at that site.

We admit that this is a limitation. As this was a pilot study, access to subjects and corresponding samples was limited to a single observation. As a result, we elected to investigate the immediate effects of therapy. Though we are aware that identifying DNA of present bacteria does not elucidate their viability, we wanted to study if there was an immediate measurable effect in order to better plan for future studies. The follow-up, full study to this pilot (currently underway) will collect longitudinal data with variable rates of time following application of SDF. We have included this as part of a larger discussion section on the sample size and timing limitations of the study. 

4. The accepted term is 16S rRNA amplicon sequencing.

We have made the changes in the manuscript.

5. The print (text figure labels) in Figures 1-3 is small and will be very difficult to read when the figures are reduced for publication.

Thank you – we will ensure that if accepted for publication this is reviewed in the proofing/typsetting stage. Additionally, in our revision following with PLOS One style guidelines, our figure labels are now embedded within the manuscript.

6. It is not clear what Fig 3 (PiCrust) contributes to the paper. None of the identified pathways are statistically significant and no meaningful conclusions can be drawn from this analysis.

Thank you for your comment. Although the data was not statistically significant due to the small sample size, nevertheless we saw a trend in the plots, specifically higher in non-responders and amongst those that are involved in caries. Additionally, Fig 3 compares the role of bacteria in some of the most important metabolic pathways of bacteria. Future studies aiming to study these pathways with a larger sample size may reveal statistically significant data. As we are actively collecting data as part of a new longitudinal study to expand on this pilot, we feel it is important to note these results.

7. The identification of Granulicatella, Leptotrrichia, and Tannerella in the non -responders is interesting, with a plausible biological basis of their roles in caries presented in the discussion.

Thank you for your helpful comments and for your review, it is sincerely appreciated.

---

## [Decision Letter · Decision Letter 1]

25 Jan 2021

Microbial population shift and metabolic characterization of silver diamine fluoride treatment failure on dental caries

PONE-D-20-34812R1

Dear Dr. Ryan Richard Ruff,

We’re pleased to inform you that your manuscript has been judged scientifically suitable for publication and will be formally accepted for publication once it meets all outstanding technical requirements.

Kind regards,

Ping Xu

Academic Editor

PLOS ONE

Additional Editor Comments (optional):

Reviewers' comments:

Reviewer's Responses to Questions

**Comments to the Author**

1. If the authors have adequately addressed your comments raised in a previous round of review and you feel that this manuscript is now acceptable for publication, you may indicate that here to bypass the “Comments to the Author” section, enter your conflict of interest statement in the “Confidential to Editor” section, and submit your "Accept" recommendation.

Reviewer #1: All comments have been addressed

Reviewer #2: All comments have been addressed

2. Is the manuscript technically sound, and do the data support the conclusions?

Reviewer #1: Yes

Reviewer #2: (No Response)

3. Has the statistical analysis been performed appropriately and rigorously? 

Reviewer #1: Yes

Reviewer #2: (No Response)

4. Have the authors made all data underlying the findings in their manuscript fully available?

Reviewer #1: Yes

Reviewer #2: (No Response)

5. Is the manuscript presented in an intelligible fashion and written in standard English?

Reviewer #1: Yes

Reviewer #2: (No Response)

6. Review Comments to the Author

Reviewer #1: I have no further comments, the authors did address all my listed concerns. Thank you for being responsive to my suggestions.

Reviewer #2: (No Response)

7. PLOS authors have the option to publish the peer review history of their article (what does this mean?). If published, this will include your full peer review and any attached files.

Reviewer #1: No

Reviewer #2: No

---

## [Editor Report · Acceptance letter]

24 Feb 2021

PONE-D-20-34812R1 

Microbial population shift and metabolic characterization of Silver Diamine Fluoride treatment failure on dental caries 

Dear Dr. Ruff:

I'm pleased to inform you that your manuscript has been deemed suitable for publication in PLOS ONE. Congratulations! Your manuscript is now with our production department. 

Kind regards, 

on behalf of

Professor Ping Xu 

Academic Editor

PLOS ONE